# Genomic landscape and clonal architecture of mouse oral squamous cell carcinomas dictate tumour ecology

Inês Sequeira[1,2], Mamunur Rashid[3,5], Inês M. Tomás[1,5], Marc J. Williams [4], Trevor A. Graham [4], David J. Adams [3], Alessandra Vigilante[1] & Fiona M. Watt [1✉]

To establish whether 4-nitroquinoline N-oxide-induced carcinogenesis mirrors the heterogeneity of human oral squamous cell carcinoma (OSCC), we have performed genomic analysis of mouse tongue lesions. The mutational signatures of human and mouse OSCC overlap extensively. Mutational burden is higher in moderate dysplasias and invasive SCCs than in hyperplasias and mild dysplasias, although mutations in *p53*, *Notch1* and *Fat1* occur in early lesions. *Laminin-α3* mutations are associated with tumour invasiveness and *Notch1* mutant tumours have an increased immune infiltrate. Computational modelling of clonal dynamics indicates that high genetic heterogeneity may be a feature of those mild dysplasias that are likely to progress to more aggressive tumours. These studies provide a foundation for exploring OSCC evolution, heterogeneity and progression.

[1] Centre for Stem Cells & Regenerative Medicine, King's College London, Guy's Hospital, Great Maze Pond, London SE1 9RT, UK. [2] Institute of Dentistry, Barts and the London School of Medicine and Dentistry, Queen Mary University of London, 4 Newark Street, London E1 2AT, UK. [3] Experimental Cancer Genetics, The Wellcome Trust Sanger Institute, Hinxton, Cambridgeshire CB10 1SA, UK. [4] Centre for Cancer Genomics and Computational Biology, Barts Cancer Institute, Queen Mary University of London, London EC1M 6BQ, UK. [5]These authors contributed equally: Mamunur Rashid and Inês M. Tomás. ✉email: Fiona.Watt@kcl.ac.uk

O ral squamous cell carcinoma (OSCC), a subset of head and neck cancer, accounts for 355,000 new cancer cases annually worldwide[1] and has a 5-year survival rate of only 50%[2]. Current approaches to analysing the genetic heterogeneity of human head and neck cancer focus primarily on the genetic lesions in frank tumours[3–7]. Recent whole-exome sequencing[3,4,8] has shown that most tumours have inactivating mutations in *Tp53*, while mutations in *Notch1*, *Fat1*, *Pik3ca* and *Casp8* are also frequent[3,8]. In addition to genetic heterogeneity, significant inter-tumour heterogeneity is observed by histopathology, reflecting the tumour site of origin, proliferation and differentiation, depth of invasion and degree of inflammation[9]. There is also evidence of field cancerisation, whereby tumours develop within large pre-neoplastic regions (fields) of clonally related cells[10].

Epithelial cells can be readily cultured from resected tumours and retain the range of mutations found in the original biopsy[11,12]. However, in vitro experimental models are unable to capture malignant transformation of early stage lesions and human tumour xenografts cannot mimic the impact of a functional immune system on tumour progression. To overcome the limitations of current approaches to studying head and neck cancer, we have focused on an autologous mouse model of OSCC that most commonly affects the tongue. Tumours are induced by chronic oral administration of the water-soluble carcinogen 4-nitroquinoline-1-oxide (4NQO)[13–15] that mimics the alterations caused by tobacco mutagens. As in human OSCC, invasive tumours are preceded by epithelial hyperplasia and dysplasia. 4NQO forms DNA adducts, causing substitution of adenosine for guanosine, and induces intracellular oxidative stress resulting in mutations and DNA strand breaks[16]. 4NQO is known to induce point mutations in *HRas* with subsequent loss of heterozygosity[17,18], upregulation of EGFR[19], *p53* mutations[20] and reduced expression of the cell cycle inhibitor *p16*[19]. These effects are similar to the genetic alterations induced by tobacco carcinogens[19,21]. However, while individual molecular changes induced by 4NQO have been documented[21–26], a systematic genomic analysis has not been performed.

Here we provide a comprehensive genomic and clonal analysis of 4NQO-induced tumours at different stages of development and correlate genetic changes with the tumour microenvironment. We have uncovered the mutational landscape of 4NQO-induced tongue squamous cell carcinoma (TSCC), identifying the recurrently mutated genes and their time of onset. We have linked *Lama3* mutations to tumour invasiveness and *Notch1* mutations to an increased immune cell infiltrate, and explored the clonal architecture of individual tumours. We conclude that the 4NQO carcinogenesis model captures many of the hallmarks of human OSCC.

## Results

**Whole-exome sequencing of 4NQO-induced TSCC.** 4NQO was delivered in the drinking water (100 µg/mL) of C57BL/l6 mice for 16 weeks. Mice were given normal drinking water for a further 10 weeks. We culled animals every 2 weeks during this period ($n = 75$) (Fig. 1a). Lesions were identified and characterised macroscopically, and formalin-fixed and paraffin-embedded (FFPE) sections were analysed by microscopy (Fig. 1d, Supplementary Fig. 1a, Supplementary Data 1 and 2). We performed whole-exome sequencing (WES) of 69 4NQO-treated samples, which were classified as follows: tongue that was histologically normal but had been exposed to 4NQO (grade 1; $n = 5$), hyperplasia (grade 2; $n = 19$), mild dysplasia (grade 3; $n = 15$), moderate dysplasia (grade 4; $n = 14$), severe dysplasia (grade 5; $n = 3$) and invasive SCC (grade 6; $n = 13$) (Fig. 1c, d). WES of four tumours failed QC, and so mutational analysis was performed on $n = 65$ tumours. Given the small number of severe

dysplasias and their histological similarity to moderate dysplasias, we combined grades 4 and 5 for subsequent analysis. Most lesions were found in the dorsal and ventral tongue, with a higher number of advanced tumours (grades 4–6) in the dorsal tongue (Fig. 1c).

For WES, we microdissected the epithelial compartment of FFPE-fixed sections and extracted DNA (Supplementary Fig. 1a). We performed WES using the HiSeq2000 Illumina platform (average depth of 105x) of tongue 4NQO-treated samples, and matched snap-frozen normal ear skin DNA from the same animals. After somatic variant calling with CaVEMan (Cancer Variants Through Expectation Maximisation)[27], germline variant filtering and exclusion of common variants (Supplementary Fig. 1a), we identified a total of 515,212 mutations (Fig. 1e) and 663 Indels in the 65 samples. Higher mutation burden was observed in moderate dysplasias and invasive SCCs than in hyperplasias and mild dysplasias (Fig. 1e). Overall, C > T mutations dominated the spectra, either in CpG or in other dinucleotides. T > C and C > A mutations were also common (Fig. 1f). The relative abundance of the different mutations did not change significantly as lesions progressed from grade 1 to grade 6 (Supplementary Fig. 1b).

The analysis was performed on FFPE samples because areas of individual tumours could be micro-dissected with more accuracy than in frozen sections, thereby reducing contamination of the epithelial compartment with adjacent stroma. It has previously been reported that formaldehyde fixation can result in artefacts; these are of low allelic frequency (<5%) and share a "C > T" mutational signature resulting from cytosine deamination[28,29]. To take into account these potential artefacts, we used an effective filtering strategy with associated empirical false-discovery estimates in our analysis pipeline; we performed rigorous quality control on extracted DNA and assessed sequence data quality, read alignments, library complexity, raw error rate, and consensus base calls. In addition, we used samples that were less than 6 months old and confirmed that they did not have the typical increase in C > T transitions observed in archival FFPE[30,31] (Supplementary Fig. 1c). Therefore analysis of the allelic frequencies of our FFPE samples confirmed that they are unlikely to be artefacts of formalin fixation[31].

DNA from frozen sections of ear skin, taken prior to 4NQO treatment, provided the controls for germline mutations in each mouse. In addition, WES was performed on FFPE sections of untreated tongues from two C57Bl/6N mice (samples MD5547b and MD5548b).

Mutagenic processes generate characteristic point mutation rate spectra (mutational signatures) that are identified as common patterns based on counts of mutations and their sequence context. We extracted 3 de novo mutational signatures from our samples, indicated as Signatures A, B and C (Fig. 1g and Supplementary Fig. 1d). Half of the more aggressive samples (grade 4–6) were enriched in Signature C, while Signature A was enriched in grade 1–3 samples. We also analysed the similarity between each mutational profile and the 30 recurrent base substitution patterns that are archived in the Catalogue of Somatic Mutations in Cancer (COSMIC)[32] (Fig. 1h and Supplementary Fig. 1e, f). Moderate and severe dysplasias and invasive SCCs (grades 4–6) showed an enrichment of signatures related to advanced human OSCC. In particular, these tumours presented a high similarity to Signatures 4, 18 and 29 (cosine of similarity > 0.90) (Fig. 1g). Signatures 4, 18 and 29 are observed specifically in human head and neck SCC, oesophageal, stomach and gingiva-buccal oral SCCs and are known to be associated with mutations due to tobacco use[33]. Comparison of the de novo signatures with COSMIC signatures revealed that de novo signature C correlates with COSMIC signature 4 while de novo signature B with COSMIC signature 5

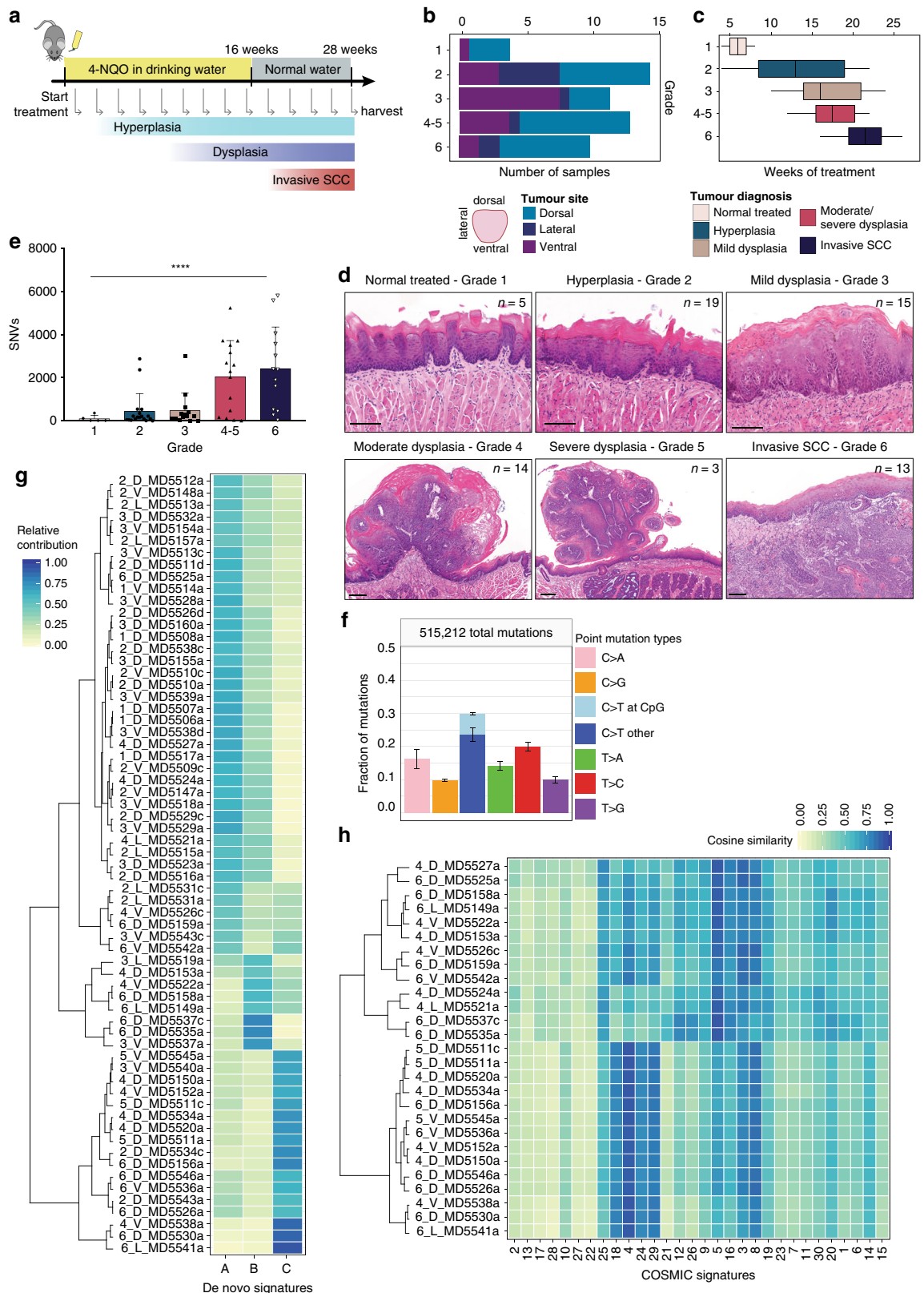

(cosine of similarity of 0.917 and 0.871, respectively) (Supplementary Fig. 1e).

**Mutated genes in 4NQO-induced TSCCs.** We next analysed all the genes that were mutated in our samples and correlated the

mutated genes with the corresponding biological information, including tumour grade, degree of differentiation and invasiveness, proliferation index and extent of immune cell infiltrate (CD45[+], CD3[+] and CD68[+] cells) (Fig. 2; Supplementary Figs. 2, 3 and 5a, d, Supplementary Data 1). A total of 14,003 genes had at least one mutation in at least one of the samples analysed

**Fig. 1 Mutational burden and signature of 4NQO-induced TSCCs. a** Schematic of 4NQO tumorigenesis protocol. Mouse image obtained from Pixabay (https://pixabay.com/service/terms/) and used in accordance with the terms of the Pixabay License. OSCCs were induced in C57BL/6 mice ($n = 75$) by administration of 4NQO carcinogen in the drinking water of mice for 16 weeks. Harvesting of tumours was done every 2 weeks for a total of 26 weeks. **b** Distribution of tumours on the tongue by grade. **c** Distribution of tumour grade according to the number of weeks of 4NQO treatment represented as boxplots (the middle line is the median, the lower and upper hinges are the first and third quartiles, the upper whisker extends from the hinge to the largest value no further than 1.5× inter-quartile range from the hinge and the lower whisker extends from the hinge to the smallest value at most 1.5× inter-quartile range of the hinge. Data beyond the end of the whiskers are outliers that are plotted individually). **d** Histological heterogeneity of 4NQO-induced TSCCs. Representative images of haematoxylin & eosin staining (H&E) of tongue lesions for each tumour grade: Grade 1 (normal 4NQO-treated, $n = 5$), Grade 2 (hyperplasia, $n = 19$), Grade 3 (mild dysplasia, $n = 15$), Grade 4 (moderate dysplasia, $n = 14$), Grade 5 (severe dysplasia, $n = 3$) and Grade 6 (invasive SCC, $n = 13$). Scale bars 100 µm. **e** Differences in mutation burden by grade. Barplot depicts total SNVs per tumour grade ($n = 65$ biologically independent samples, bar graphs represent mean ± SD, ****$p < 0.0001$, one-way ANOVA). **f** Mutational signature of all 4NQO-induced OSCCs ($N = 515,212$ mutations, $n = 65$ samples, bar graphs represent mean ± SD over all samples). **g** De novo signatures. **h** Similarity heatmap featuring the contribution of known COSMIC signatures calculated using *MutationalPatterns* (R package) for tumour grades 4, 5 and 6. Each number corresponds to the tumour reference: tumour grade (1–6), tumour site (D, dorsal; L, lateral; V, ventral) and the tumour sequencing ID.

(Supplementary Data 2). Overall, the total number of single nucleotide variants (SNVs) increased with increasing tumour grade (Fig. 1e, Supplementary Fig. 3). However, a subset of hyperplastic lesions had an increased number of SNVs, correlating with the length of 4NQO treatment (treatment longer >12 weeks, Fig. 2, Supplementary Fig. 3).

Among the top 100 mutated genes (Supplementary Fig. 2) were several that are known to be mutated in human head and neck SCC, including *Trp53*, *Erbb4* and 4 genes of the *Fat* family. We then focused on genes that are known to harbour bona fide driver mutations in human TSCCs[34] (see Methods and Supplementary Data 3). From the mouse tongue recurrent mutated genes (all SNVs, including missense, nonsense, ess-splice and indels), the highest ranked genes included *Trp53*, *Fat1*, *Syne2* and *Notch1* (Fig. 2).

*Fat1* was mutated in 30% of the tumours, particularly in moderate and severe dysplasias and invasive SCC (Fig. 2). Loss of *Fat1* has been variously associated with improved or decreased patient survival[35,36] and gives a competitive advantage to OSCC cells in culture[12]. *Trp53* is a well-known tumour suppressor gene[37]. In human head and neck SCC, the overall frequency of *Tp53* mutations is approximately 70%[3,4,6] but the incidence in TSCCs is lower at 38.3%[34]. Consistent with the human data, *Tp53* was mutated in 30% of our samples and mutations were found in more aggressive lesions (dysplasias and invasive SCC), suggesting that this gene plays an important role in tumour progression. *Trp53* mutations were also found in hyperplastic lesions but only in mice treated for a long period with 4NQO (Fig. 2, Supplementary Fig. 3), supporting the observation that loss of *p53* alone is not responsible for tumour initiation[38]. *Syne2* and *Notch1* were mutated in 23 and 9% of our samples, respectively and, like *Fat1* and *Trp53*, are frequently mutated in human head and neck SCC[3–7].

To detect additional cancer driver genes (i.e. genes under positive selection in cancer), we used the dNdScv method to calculate dN/dS ratio (the normalised ratio of non-synonymous to synonymous mutations) for missense, nonsense and essential splice mutations[39]. This revealed 35 genes under positive selection ($p$-value <0.001) (Fig. 2, Supplementary Data 4). These data show similarity between the recurrently mutated genes in human OSCC and mouse OSCCs and identify additional possible driver genes such as *Chuk*, a serine/threonine protein kinase that is involved in nuclear factor κ-B (NF-κB) activation and functions as a tumour suppressor by controlling turnover of cyclin D1[40,41]. A previous study has shown that genes involved in the NF-κB pathway, including *Chuk*, are upregulated in skin SCCs[42]. In OSCC, Chuk has been previously associated with ELAVL1[43], an RNA-binding protein that stabilises the transcripts of many cancer-related genes and is also a putative driver gene[44] (Supplementary Data 4).

*Kras*, a gene within the *RAS* family involved in signal transduction and cell cycle regulation, was also mutated in 12% of our samples (Fig. 2).

Antibody labelling supported the identification of mutations in two of the genes (Fig. 3a–d). Tumours that were mutant for *Lama3* had areas in which the protein was absent from the basement membrane, consistent with a loss of function mutation (Fig. 3a, b), leading to multifocal breaks of the epithelial-mesenchymal boundary. *Lama3* mutations were only observed in moderate and severe dysplasias and invasive SCC (Fig. 3a, Supplementary Fig. 2), suggesting that loss of *Lama3* expression may contribute to a more invasive phenotype[45]. Tumours mutated in *p53* exhibited enhanced staining with anti-p53 (Fig. 3c, d) suggesting they had stabilising mutations in *Trp53*. To gain a broader perspective on changes in the tumour ecosystem[46], we performed Gene Ontology (GO term) analysis based on the driver genes we identified (Fig. 3c; Supplementary Data 5). In addition to changes in the epithelial compartment (GO terms: morphogenesis, anatomical structure), GO term analysis predicted alterations in cytokine-mediated signalling and immune infiltrate.

**Highly mutated hyperplasias lack cancer driver gene mutations.** Although in general the number of SNVs increased with increasing tumour grade (Fig. 2), we observed a subgroup of hyperplastic lesions with a high number of SNVs. These corresponded to lesions in tongues that had been treated with 4NQO for prolonged periods (>10 weeks of 4NQO treatment) without developing into aggressive tumours. Combining transcriptome (RNAseq data from normal tongue[23]) and mutational analysis (Fig. 4), we analysed the expression level (log CPM – Count Per Million normalised read count) of genes mutated in these hyperplastic lesions (Fig. 4a). We placed the genes into three categories based on the average CPM values: (i) low expression (log CPM < 4.5); (ii) medium expression (log CPM between 4.5–10); (iii) high expression (log CPM > 10). The number of genes in each category is shown in Fig. 4b, c.

KEGG (Kyoto Encyclopedia of Genes and Genomes) pathway analysis of genes with medium and high expression levels (Fig. 4d) in both the hyperplasias and invasive SCC showed significant enrichment in cancer associated pathways: "Focal adhesion", "Proteoglycans in cancer" and "Axon guidance" (genes related to cell migration and apoptosis). Nevertheless, a minority of genes mutated in hyperplastic lesions only (243 out of 1691) were enriched in pathways related to cancer progression ("ECM-receptor interaction", "Calcium signalling"), whereas all the genes mutated in invasive SCC and not in hyperplasias were enriched in pathways associated with cancer[47]. These results suggest that the reason why the hyperplasias with a high number of SNVs do not

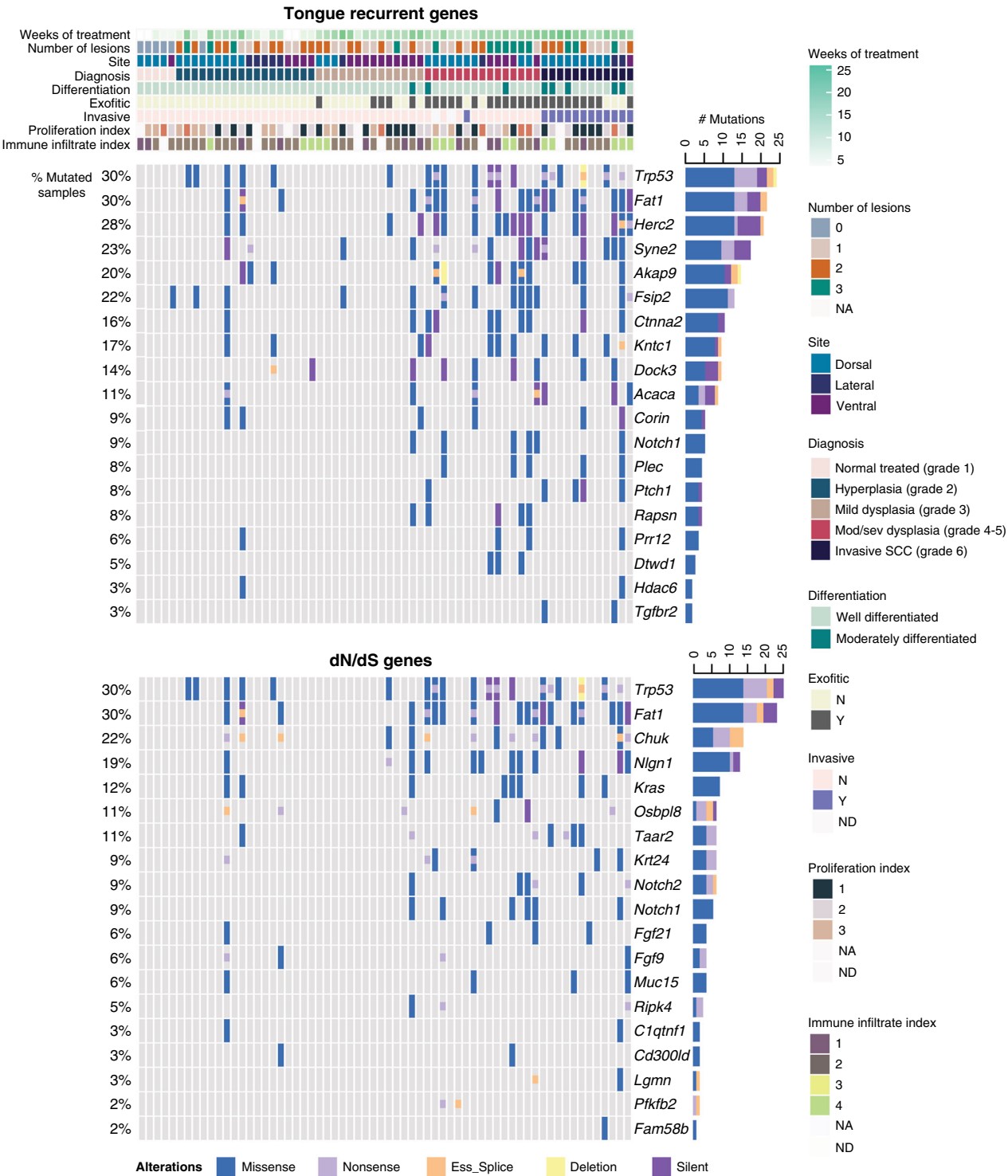

**Fig. 2 Data matrix showing mutational landscape of 4NQO-induced TSCCs compared with tumour ecology, including recurrent somatic mutations.** The top panel shows the key clinical parameters: colour coding indicates number of weeks of treatment, number of lesions per tongue, site of the sequenced lesion (dorsal, ventral or lateral), diagnosis (tumour-grade), differentiation status, exophytic tumour (yes/no), invasive tumour (yes/no), proliferation index, and immune-infiltrate index (CD45⁺ cells) (Supplementary Data 1 2 and 3). The middle and the bottom panels show the significantly mutated genes (rows; tongue recurrently mutated genes and a subset of dNdS genes, respectively) coloured by the types of mutation. Samples (columns, n = 65) are organised by diagnosis by increasing grade. Somatic mutations are presented according to the type of mutation (missense, nonsense, exonic splicing silencer 'Ess_Splice', deletion or silent mutations; colours defined in key). All mutations are listed in Supplementary Data 3, 4 and 5. The distribution between the types of somatic alterations for each gene is shown in the histogram (right). Percentages on the left represent the fraction of tumours harbouring mutations in the corresponding gene.

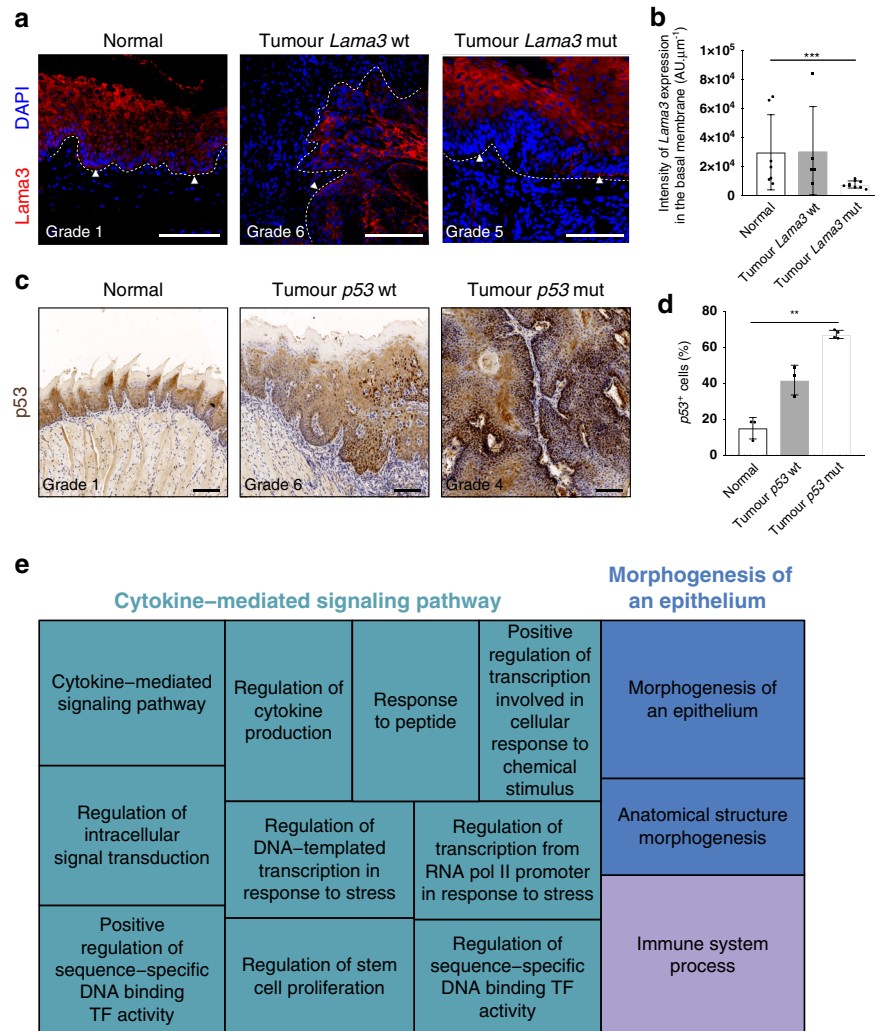

**Fig. 3 Expression of Lamininα3 and p53 in mutated tumours. a**, **c** Representative images of immunostaining of *Lamininα3* (**a**) and *p53* (**c**) in normal tissue, *wild-type* and mutated tumours. Arrowheads indicate areas with loss of *Lamininα3* expression in mutated tumours ($n = 6$ lesions mutated for *Lamininα3*). Samples were counterstained with nuclear dye DAPI (4′,6-diamidino-2-phenylindole) (**a**) or with haematoxylin (**c**). Dotted line delineates basement membrane. Scale bars, 100 μm. **b** Quantification of the expression of *Lamininα3* in the basement membrane ($n = 3$ normal samples with $n = 7$ regions of interest (ROIs) quantified, $n = 3$ tumour *Lama3 wt* samples with $n = 5$ ROIs quantified, $n = 4$ tumour *Lama3* mutated samples with $n = 9$ ROIs quantified). **d** Quantification of *p53* positive cells ($n = 3$ normal, 3 tumour *p53 wt* samples, 4 tumour *p53* mutated samples). Bar graphs represent mean ± SD, **$p \leq$ 0.05, ***$p \leq 0.001$, $p = 0.0014$ for *p53*, $p = 0.0010$ for *Lamininα3*, one-way ANOVA, Kruskal–Wallis test. **e** Gene Ontology analysis of driver genes. Each rectangle is a single cluster representative of GO categories (biological processes), joined together into three main categories (superclusters): cytokine-mediated signalling pathway, morphogenesis of an epithelium and immune system process. Size of the rectangles is adjusted to reflect the frequency of the GO term in the underlying GO analysis database.

progress to invasive SCC is because the mutated genes are not essential for tumour progression.

**Correlation between mutational landscape and tumour ecology.** To determine whether the mutational landscape of individual tumours correlated with tumour phenotype, we quantified the immune cell infiltrate by labelling sections with antibodies against CD45 (all immune cells), CD3 (T-cells) and CD68 (macrophages) and measured the proportion of proliferating cells (EdU+) in all sequenced tumours (Fig. 5a–d; Supplementary Fig. 4a, b, Supplementary Table 1 and Supplementary Data 1). There was considerable heterogeneity within each grade and there was no statistically significant difference between different grades.

We next performed linear regression analysis to compare multiple biological parameters (linearMod, pvalue cut-off < 0.05) (Supplementary Data 1) with the frequency of mutations and

Variant Allele Frequency (VAF) in individual genes (Fig. 5e; Supplementary Fig. 4c, d, Supplementary Data 6). The total number of mutant genes was significantly correlated with exofitic tumours (i.e., moderate and severe dysplasias), aggressive lesions (grades 4–6) and proliferation index (Supplementary Fig. 4d). Although there was no significant difference in the number of proliferative basal cells in different grades of lesion (Fig. 5d) there was an increased abundance of proliferative Krt14+ cells above the basal layer (Supplementary Fig. 4b). In addition, although there was no significant difference in proliferation across all tumour grades, there was a significant increase in SCC compared to hyperplasias ($p = 0.0262$, Fig. 5d).

*Lama3* mutations correlated with tumour grade ($p = 0.00233$), total immune infiltrate (CD45+, $p = 0.029824$), exofitic tumour growth ($p = 0.05172$) and tumour invasiveness ($p = 0.02748$) (Fig. 5e, Supplementary Data 6). Mutations in *Fat1* correlated positively with proliferation ($p = 0.04094$) and inversely with

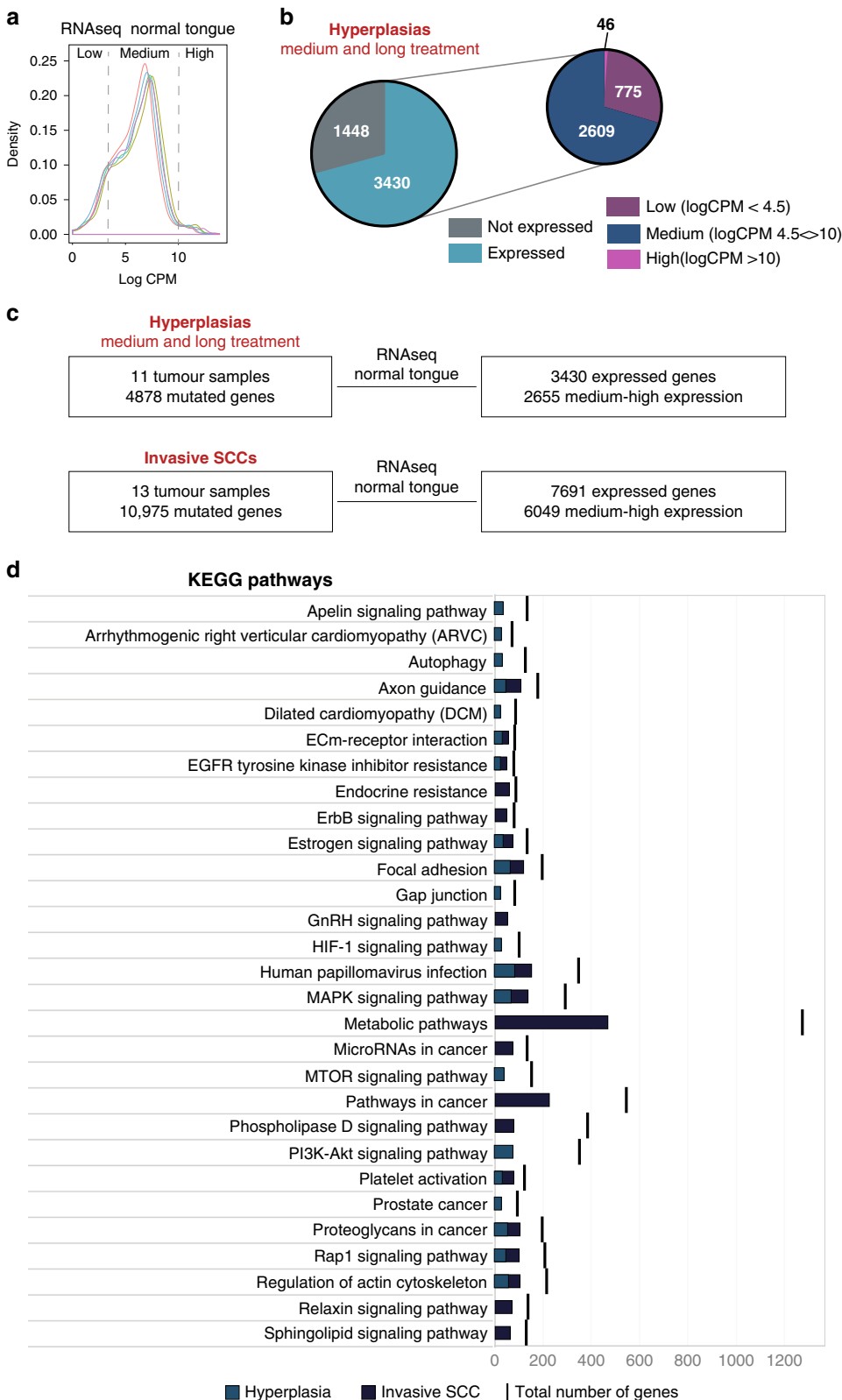

**Fig. 4 Functional analysis of the mutations in medium-long treated hyperplasia. a** Expression profile (log CPM – Count Per Million normalised read count) of the 5 datasets of normal tongue samples[23]. Graph represents the average of CPM values; x-axis is divided in 3 categories: low, medium and high expression genes. **b** Pie-chart with the number of expressed genes in each category of genes mutated in hyperplastic samples. **c** Total number of samples and mutated genes in medium and long-treated hyperplasias and in invasive SCC before (left panels) and after (right panels) filtering for the genes medium to highly expressed in the tongue. **d** Top 10 KEGG pathways enriched in tongue expressed genes mutated in medium and long treated hyperplasias and in invasive SCC. The number of mutated genes in the pathway is indicated as well as the total number of genes in the pathway.

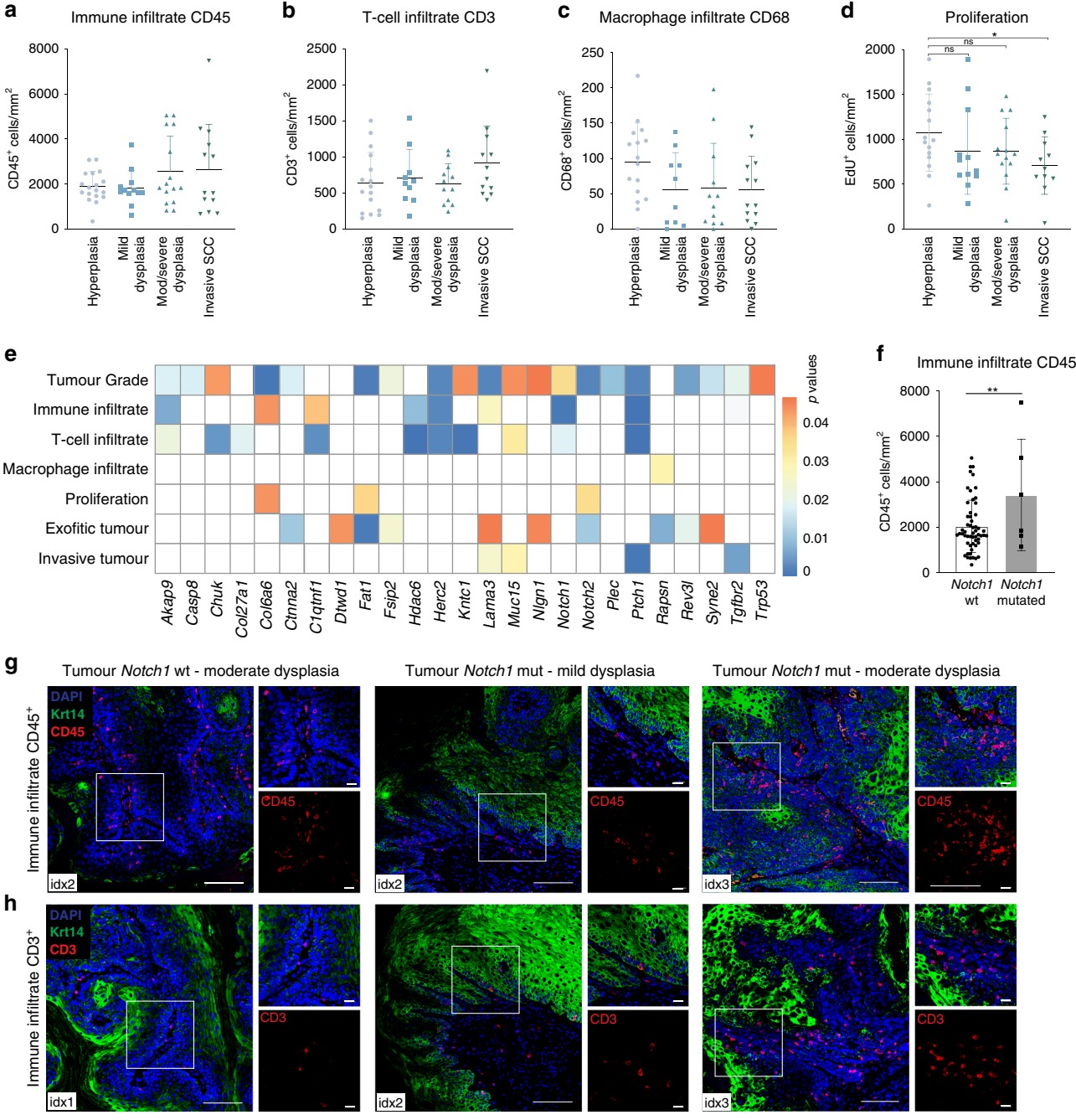

**Fig. 5 Correlation between mutational landscape and tumour ecology. a–c** Quantification of immune infiltrate (CD45+ cells/mm²), T-cell infiltrate (CD3+ cells/mm²) and macrophage infiltrate (CD68+ cells/mm²) in the different tumour grades (hyperplasia $n = 18$, mild dysplasia $n = 12$, moderate/severe dysplasia $n = 16$, invasive SCC $n = 12$ mice, 2 sections/mouse and >6 fields quantified per section; means ± SD are shown, non-significant, one-way ANOVA). **d** Quantification of proliferation (EdU+ cells/mm² in the different tumour grades (hyperplasia $n = 15$, mild dysplasia $n = 12$, moderate/severe dysplasia $n = 14$, invasive SCC $n = 11$, 2 sections/mouse and >6 fields quantified per section; means ± SD are shown. Group comparison with one-way ANOVA, non-significant. Multiple comparison: unpaired $t$-tests between each grade, non-significant for all except between invasive SCC and hyperplasia, *$p < 0.05$, $p = 0.0262$. **e** Linear regression analysis to establish the relationship between the total number of mutations in each gene (columns) and the clinical parameters (rows). The heatmap shows the $p$-value for each correlation (Supplementary Data 7). **f** Comparison of immune infiltrate (CD45+/mm²) between *Notch1* mutated and *Notch1* non-mutated (wt) tumours ($n = 56$ *Notch1* wild-type lesions and $n = 6$ *Notch1* mutated lesions. Bar graphs represent mean ± SD, *$p < 0.05$, $p = 0.0083$, unpaired one-tailed $t$-test). **g, h** Quantification of immune infiltrate of CD45+ cells (**g**) and T-cell infiltrate (CD3+ cells) (**h**) in *Notch1* mutated tumours. Immunofluorescence staining of representative sections of *Notch1* wt and *Notch1* mutated tumours labelled with anti-krt14 (green) and anti-CD45 or anti-CD3 (red), counterstained with nuclear dye DAPI. Immune infiltrate was quantified by the number of CD45+ or CD3+ cells per mm² (Supplementary Fig. 4a, b). All tumours were analysed and for each tumour ≥6 regions were quantified per section ($n = 2$–3 sections/tumour/staining). **g** idx2, CD45+ immune infiltrate index 2 (2000–5000 cells/mm²); idx3, index 3 (5000–10,000 cells/mm²); idx4, index 4 (>10,000 cells/mm²). **h** idx1, CD3+ immune infiltrate index 1 (<500cells/mm²); idx2, index 2 (500–1000 cells/mm²); idx3, index 3 (>1000 cells/mm²). Scale bars 100 μm, 20 μm in magnified squares.

tumour grade[35] ($p = 0.00128$) and exofitic tumour growth ($p = 0.02702$)(Fig. 5e). *Trp53* mutations positively correlated with high grade tumours ($p = 0.05160$) but no other parameters, while *Notch1* mutations correlated significantly with an increased total immune cell infiltrate (CD45[+], $p = 0.00069$), T-cell infiltrate (CD3[+], $p = 0.02019$) and high tumour grade ($p = 0.01288$)(Fig. 5e, Supplementary Data 6). We validated these observations by confirming that the tumours with *Notch1* mutations had a higher stromal CD45[+] and CD3[+] infiltrate than tumours that were wild type for *Notch1* (Fig. 5f–h). There was no correlation between *Notch1* mutational status and macrophage infiltration (CD68[+] cells) (Fig. 5e).

**Tumour clonal dynamics.** The somatic mutation theory of cancer invokes a clonal origin of tumours, with lesions arising from sequential mutations in the progeny of a single cell[48]. However, it is now apparent that genetically distinct clonal sub-populations can co-exist within a single tumour for long periods of time[49–52]. To investigate the clonal dynamics and architecture of 4NQO-induced TSCCs, we first analysed mice treated with 4NQO that had more than one distinct lesion at the time of harvest. In 8 mice analysed the lesions were at different stages of progression, with hyperplasia, dysplasia and/or SCC on different locations of the tongue (Supplementary Fig. 5). In 4 out of 8 animals, the different lesions had no common mutations and in 2 out of 8 there were ≤3 mutations in common. Of the remaining two mice, only one had a significant number of mutations shared between two lesions ($n = 1189$, Supplementary Fig. 5a). In this mouse both lesions were located in close proximity in the dorsal region of the tongue (Supplementary Fig. 5b, 2_D_MD5534c and 4_D_MD5534a). These observations suggest that lesions in different regions of the same tongue are not clonally related, except when located in close proximity to one another.

We next assessed the clonality of mutations within individual lesions using computational methods of clonal selection, analysing the variant allele frequency (VAF) of each mutation[53–55]. A detailed analysis of subclonality was precluded in most cases due to the low median VAF (0.11). This may be due to low tumour purity or the presence of multiple low frequency clones. In some of the higher-grade lesions, we did however observe enrichment in the number of clonal mutations (VAF > 0.25) together with an enrichment of high frequency putative driver mutations (Fig. 6c, sample 6_L_MD5541, invasive SCC; Fig. 6b, sample 4_V_MD5538A, moderate/severe dysplasia). This may be due to the clonal expansion of a fitter clone carrying multiple drivers.

There were other noticeable differences in VAF distributions across samples that reflect the evolutionary history of different lesions. Principally, some samples had a large number of clonal mutations whereas other samples had very few (Fig. 6b, c). The number of clonal mutations is a measure of the mutation burden of the most recent common ancestor (MRCA) of the sampled population. A high clonal mutation burden may therefore be the result of a selective sweep of a fitter clone, the larger the mutation burden the later the sweep and consequently the fitter the clone[55]. Consistent with this hypothesis, when we compared tumour grade with tumour clonal selection, most of the mild dysplasias had few clonal mutations, while a subset of higher-grade tumours presented patterns consistent with clonal mutations (Fig. 6c). However, we observed that tumours bearing subclonal mutations started to emerge as early as grade 3, mild dysplasia (Fig. 6c).

To broadly assess which mutated genes were present in each clone, we annotated the mutated genes in each VAF peak. We observed that in high-grade tumours bearing a clone, *Notch1*, *Fat1* and *Syne2* mutations had a similar VAF (Fig. 6c), whereas in lower grade lesions subclonal mutations in *KRAS* (Fig. 6b) and

*Tp53* (Fig. 6c) were observed. We speculate that high genetic heterogeneity may be a feature of those mild dysplasias that are likely to progress to more aggressive tumours. Taken together, these data shine light on the evolutionary dynamics of tumour progression.

**Discussion**

We have found that the mutational spectrum of human OSCC is faithfully recapitulated in 4NQO-induced mouse tumours, including mutations in *Trp53*, *Pik3ca*, *Notch1*, *Fat1, Lama3* and *Syne2*. In addition, we provide a comprehensive analysis of the genomic landscape of 4NQO carcinogenesis at different stages of tumour development, from early hyperplasia to invasive SCC. This provides insights that cannot be gained from end-point analysis of malignant tumours.

By correlating the different biological parameters of individual tumours with their mutational spectrum, we were able to show that mutant *Fat1* is associated with tumour grade and high proliferation rate, while *Trp53* correlates with tumour grade, and *Notch1* with immune infiltrate (Fig. 5e–h). *Notch1* is known to regulate the crosstalk between different cells in the tumour microenvironment[56]. The tumour-promoting effect of *Notch1* inactivation in the epithelia of the skin, oral cavity and oesophagus is well established, and our data suggest that this is mediated, at least in part, by triggering inflammation[8,57–62]. In colorectal cancer, *Notch1* signalling leads to highly penetrant metastasis through control of inflammatory chemokine expression[63]. Furthermore, a recent report on human head and neck SCC has shown how mutations in multiple genes converge on the Notch signalling pathway, making NOTCH inactivation a hallmark of this cancer[64].

The loss of *Laminin-α3* (*Lama3*) in higher-grade tumours (moderate dysplasia to invasive SCC) (Fig. 3a, c, Fig. 6a, Supplementary Fig. 2) could be used as a marker to predict metastatic behaviour. Our results provide grounds for optimism that in future it may be possible to accurately predict mutational burden from analysis of tumour histology, leading to new and earlier therapeutic interventions. Since several of the mutant genes, such as *Lama3* and *Trp53*, are readily detected by antibody staining (Fig. 3a, b), it may also be possible to map the ecology of individual tumours with respect to the appearance of specific mutations.

Clonal architectures are prevalent across cancer types[65] and it is now apparent that genetically distinct clonal subpopulations can co-exist within a single tumour for long periods of time[49–51]. Passenger mutations that are phenotypically silent in normal tissue can confer a growth advantage in the context of a tumour[54] and increase invasiveness[12,66]. Understanding the dynamics of cancer evolution can provide a quantitative measurement of tumour heterogeneity from early stage-lesions to invasive SCCs. By inferring tumour clonality from mutational burden it is possible to predict the evolutionary outcome of an early lesion[52].

We detected the emergence of clonal organisation of TSCCs from early stage lesions (mild dysplasia) to moderate/severe dysplasia and invasive SCC (Fig. 6). A subset of higher-grade tumours presented patterns consistent with clonal mutations, consistent with previous observations that clonal diversity encompasses the genetic heterogeneity of more aggressive tumours. However, in addition, heterogeneity was a feature of some mild dysplasias, which we speculate are likely to progress. Their identification could improve early interventional treatment strategies, since genetic heterogeneity is likely to be one of the main causes for treatment failure.

In summary, we have been able to integrate genetic and phenotypic heterogeneity[46] within a mouse model of OSCC that faithfully recapitulates many aspects of the human disease. We can

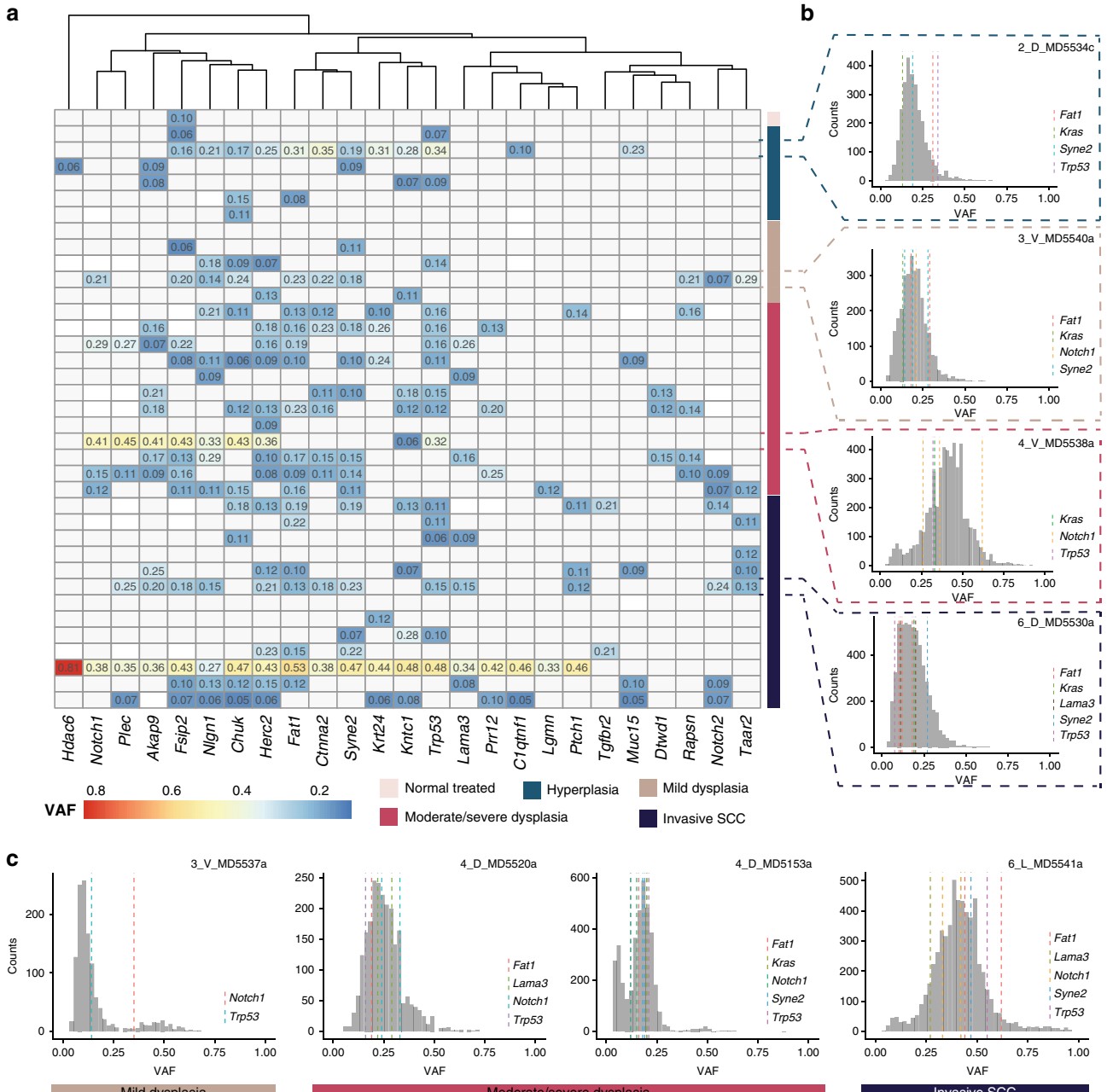

**Fig. 6 Tumour clonality. a** Heatmap showing the average VAF value of mutations of each gene (columns) in the samples where at least one mutation was detected (rows). The genes are ordered according to hierarchical clustering, and samples are ordered according to diagnosis (see colour code). **b** For each diagnosis, the distribution of the VAF of somatic mutations in one sample is shown. Grey bars represent the number of mutations for each gene. **c** Representative plots of VAF distribution of somatic mutations for higher-grade samples annotated with driver mutations. Mutations with relatively high frequency ($f > 0.25$) are likely to be clonal (public).

now build on these studies to improve diagnosis and treatment of human OSCC.

## Methods

**4NQO carcinogenesis**. All animal procedures were subject to institutional ethical review and were approved by the UK Home Office (in accordance with UK law, Animals Scientific Procedures Act 1986) at King's College London prior to commencement. Mice from both genders were maintained on the C57BL/6 N genetic background and were housed under a 12 hour light/12 hour dark cycle, at temperatures of 20–24 °C with 45–65% humidity. 4NQO (Sigma, diluted to 100 μg/mL) was administered in the drinking water. 4-NQO–containing water was prepared and changed once a week for 16 weeks. After that period, mice were given drinking water without 4NQO. Mice were maintained with regular mouse chow and water (±4NQO) *ad libitum*. Once a week, 4NQO-treated mice were sedated with inhaled

isoflurane and the oral cavities were screened for lesions (hyperplasias, dysplasias and SCCs). Mice were injected with 500 μg EdU (5-ethynyl-2'-deoxyuridine, Invitrogen) in PBS intraperitoneally 2 h before culling to assess proliferation. Sample sizes were determined on the basis of prior power calculations.

**Tumour harvesting and histology**. Tongue tissues and lymph nodes were harvested at different time-points during 4NQO treatment. Ear skin was collected as a control for sequencing, before the start of the 4NQO treatment. WES was also performed on untreated tongues from C57BL/6 N mice following the same protocol (Supplementary Fig. 1) to provide germline DNA for the elimination of single-nucleotide polymorphisms. For frozen sections, tissues were embedded in OCT (optimal cutting temperature compound, VWR), sectioned and post-fixed in 4% paraformaldehyde/PBS pH 7.4 for 10 min before staining. For paraffin sections, tongue samples were fixed with 10% neutral buffered formalin overnight before paraffin embedding. The tissues were sectioned and stained with haematoxylin and

eosin (H&E) by conventional methods. Images were acquired using a Hamamatsu slide scanner and analysed using NanoZoomer software (Hamamatsu).

**Histological classification**. Tumour grading was assessed according to the presence of the following criteria: tumour cell crowding, degree of keratinisation, exofitic or invasive growth, scattered mitotic figures and nuclear atypia[67] (Supplementary Table 1). All histology was called by a pathologist blinded to the study groups and 4NQO treatment conditions.

**Whole-exome sequencing**. Tongue lesions and normal tissue were microdissected from 20–25 FFPE sections of 5µm-thickness (Supplementary Fig. 1a). Matched control ear samples were frozen prior to DNA extraction. DNA was extracted and purified using the QIAamp DNA FFPE Tissue Kit (Qiagen) according to the manufacturer's recommendations. Exonic DNA was captured using the Agilent whole exome capture kit (SureSelect Mouse All Exon). Captured material was indexed and sequenced on the HiSeq2000 Illumina platform at the Wellcome Trust Sanger Institute at an average depth of 105x. Raw pair end sequencing reads were aligned with BWA-mem to the GRCm38 mouse reference genome[68]. Duplicated reads were marked using biobambam[69]. The sequencing metrics of the individual samples are summarised in Supplementary Data 2.

**Somatic variant detection**. Somatic variants were detected using *CaVEMan*, an expectation maximisation–based somatic substitution detection algorithm[27]. Candidate somatic variants were then filtered for quality and to remove known mouse genome variations[70]. Single point mutations overlapping known structural variants in any of the 17 mouse genomes were also removed due to high misalignment rate in these regions. Small insertion and deletion (indel) detection was performed using the *cgp-pindel* pipeline (v0.2.4w)[71]. Detected indels were then filtered for quality, sequence coverage in both tumour and normal, strand bias and for overlap with known simple repeats or indels in the in-house normal panel.

**Validation of SNVs**. All potential recurrent mutations underwent extensive manual review using the Integrative Genomics Viewer[72] (IGV v2.3) to exclude the possibility that they represented germline SNPs and to ensure models were generated using high-confidence mutations.

**Variant quality control for FFPE artefacts**. Formalin fixation of tumour biopsies can have a detrimental impact on DNA integrity and introduce $C > T/G > A$ sequencing artefacts, which are more frequently observed at a 0.01–0.10 mutant allele fraction (MAF)[31]. To reduce the potential impact of these artefacts in our FFPE samples, we tested for low allelic fraction mutations (mutant allele fraction <0.10 and read depth <10) and removed them from the signature delineation process.

**Mutation spectra analysis**. SNVs in all tumours were annotated by the 96 possible trinucleotide context substitutions (6 types of substitutions × 4 possible flanking 5′ bases × 4 possible flanking 3′ bases) and summed in each tumour. Mutational Pattern analysis (Fig. 1 and Supplementary Fig. 1) was performed using the R package *MutationalPatterns*[73]. Vcf files were used as input and the base substitutions were retrieved using the *mutations_from_vcf()* function. Three new de novo signatures were extracted using the *extract_signatures()* function. To find optimal contribution of known signatures, 30 human cancer signatures (COSMIC signatures as defined in Alexandrov et al.[32]) were downloaded from the website and the similarity between the mutational profiles of our samples and COSMIC signatures was calculated using the *cos_sim_matrix()* and visualised using the *plot_cosine_heatmap()* functions.

**Similarity between mouse SCC and human SCC data**. A list of 465 human genes from Vettore *et al.*[34] was used to compare the human *versus* mouse mutational landscape of OTSCC. Genes were divided into four classes: (i) Tongue WES – Recurrence; (ii) Tongue WES – Non-recurrence; (iii) Cancer genes, and (iv) Genes from related studies (Supplementary Data 3).

**Analysis of positively selected driver genes in 4NQO samples**. Driver genes were selected using the *dNdScv* algorithm[39] implemented in the dNdScv R package, a suite of maximum-likelihood dN/dS methods designed to quantify selection in cancer and somatic evolution. The algorithm was applied to the list of detected mutations in each sample and was used quantify dN/dS ratios for missense, nonsense and essential splice mutations using trinucleotide context-dependent substitution models to avoid common mutation biases affecting dN/dS[39,74]. The necessary reference file for the mouse genome was generated and used as input. The algorithm outputs two tables (Supplementary Data 4): the first table contains all the genes with *p* values <0.01 (obtained by Likelihood-Ratio Tests)[39] while the second one is an annotated table of coding mutations.

**Gene Ontology analysis of driver genes**. The list of genes detected with *dNdScv* algorithm was used as input for a Gene Ontology (GO) analysis using the Gorilla

(Gene Ontology enRIchment anaLysis and visualisation tool) online software (http://cbl-gorilla.cs.technion.ac.il/). Results for Biological Processes were selected and a *p*-value threshold of 0.01 applied. The visualization was obtained using the REVIGO online tool (http://revigo.irb.hr/).

**KEGG pathway analysis of mutated gene expression**. Expression data from Tang et al. (GSE54246) was downloaded from the iDEP-reads database[23] and row counts were normalised to Count Per Millions (CPM) using a custom R script (https://www.ncbi.nlm.nih.gov/geo/query/acc.cgi?acc=GSE54246). 5 samples of normal mouse tongues were used to analyse significantly altered pathways in genes mutated in hyperplastic lesions vs invasive SCC lesions. Genes were classified into 4 categories according to expression values: (i) no expression, (ii) low expression (logCPM <4.5), (iii) medium expression (logCPM 4.5 < >10), (iv) high expression (logCPM>10). We performed Kyoto Encyclopaedia of Genes and Genomes (KEGG) pathway analysis of the medium and highly expressed genes using ShinyGO software[75], selecting only the top 10 enriched pathways with a *P*-value cutoff (FDR) of 0.05.

**Tumour clonal analysis from bulk sequencing data**. To interrogate the clonality of the samples we calculated the variant allele frequency (VAF) of each mutation, only considering mutations with a depth >20.

**Generation of plots**. All plots were created using the statistical computing language R (http://www.R-project.org/). The *ComplexHeatmap R Package* (https://jokergoo.github.io/ComplexHeatmap-reference/book/) was used to create the plots in Fig. 2 and Supplementary Fig. 2 with the *oncoPrint()* and *HeatmapAnnotation()* functions to generate the heatmap and the annotation bars respectively. All other plots were made using the ggplot2 package (http://ggplot2.org/).

**Immunofluorescence staining**. Frozen sections were fixed in 2% paraformaldehyde/PBS pH 7.4 and blocked with 10% foetal bovine serum, 2% BSA, 0.02% fish skin gelatin, 0.05% TritonX100 (Sigma) and 0.05% Tween (Sigma) in PBS for 1 h at room temperature. Paraffin sections were subjected to heat-mediated antigen retrieval (citrate buffer, pH6) prior to blocking. Primary antibodies were incubated overnight at 4 °C, followed by 1 h incubation at room temperature in secondary antibodies.

The following primary antibodies were used: rabbit anti-Krt14 (Covance, PRB-155P, 1/5000), chicken anti-krt14 (Biolegend, clone Poly9060, 1/5000), rat anti-CD45 (BD Pharmingen clone 30-F11, 1/200), rabbit anti-CD3 (abcam clone SP7, 1/150), rat anti-CD68 (abcam, clone FA-11, 1/75), polyclonal rabbit anti-laminin-α3 (gift from Matthew Caley, clone R14, 1/200), rabbit anti-p53 (Leica Biosystems, Novocastra NCL-L-p53-CM5p, 1/500). For immunofluorescence, the following conjugated secondary antibodies were used: AlexaFluor488 goat anti-chicken (#A11039), AlexaFluor555 donkey anti-rabbit (#A31572), AlexaFluor594 goat anti-rat (#A11007), AlexaFluor647 chicken anti-rat (#A21472) (all Invitrogen, purchased from ThermoFisher Scientific, 1/300). EdU staining was performed with a Click-it EdU imaging kit (Life Technologies) according to the manufacturer's recommendations. DAPI (Life Technologies) was used as a nuclear counterstain. Slides were mounted using ProLong Gold anti-fade reagent (Life Technologies). Images were acquired with a Nikon A1 Upright and with a Leica SP8 Confocal microscope. Images were analysed using ICY software[76].

Rabbit anti-p53 immunohistochemistry staining was detected using a standard DAB kit (Imm PRESS™ Duet Double Staining Polymer Kit, MP-7714, anti-rabbit IgG Peroxidase ImmPACT™ DAB EqV and ImmPACT™ DAB Peroxidase Substrate Kit SK-4105, Vector Laboratories). Nuclei were counterstained with haematoxylin. Slides were mounted using DPX mounting medium. Images were acquired using a Hamamatsu Slide Scanner at 40x amplification and analysed using NanoZoomer software (Hamamatsu).

**Quantification of immunostaining**. To quantify cell proliferation, immune infiltrate, p53 and Laminin-α3 staining, images were analysed using open-source ICY software[76] plug-in Manual Counting and Spot Detector. Positively stained cells for CD45, CD3, CD68 and EdU were counted per area of tumour or per area of stroma (at least 2 sections/tumour, 3–6 regions/section) (Supplementary Fig. 5b). The total number of nuclei was quantified using DAPI staining. Laminin α3 expression quantification was performed using CellProfiler.

**Statistical analysis**. All graphs and statistical calculations were generated using R software (R version 3.5.1) and Prism8 (GraphPad) software. Statistical significance was computed with the test indicated in each figure legend. The number of animals analysed in each group is indicated in each figure.

**Reporting summary**. Further information on research design is available in the Nature Research Reporting Summary linked to this article.

## Data availability

The data from mouse TSCC whole-exome sequencing have been accessioned in the European Nucleotide Archive (ENA) under the study accession number PRJEB32924

(https://www.ebi.ac.uk/ena/browser/view/PRJEB32924). The detailed information of lesion grading and matched samples is listed in Supplementary Data 1. Publicly available databases or resources used in this research are as follows: COSMIC (http://cancer.sanger.ac.uk/cancergenome/projects/cosmic/) and Gene Ontology (https://www.ncbi.nlm.nih.gov/geo/query/acc.cgi?acc=GSE54246)[23]. The datasets used to analyse recurrent mutated genes in human TSCCs are listed in Supplementary Data 4 (https://www.ebi.ac.uk/ega/studies/EGAS00001001329)[34]. All other remaining data are available within the Article and Supplementary Files and or available from the authors upon request. A reporting summary for this article is available as a Supplementary Information file.

## Code availability

All relevant codes are available from the cited references or available from the authors upon request. Custom code was generated using R to analyse the data and to generate the plots. Code used in this article comprises: (i) CaVEMan, (ii) custom code for filtering variants, (iii) cpg-pindel pipeline (v0.2.4w) for indel detection, (iv) MutationalPattern script, (v) dndscv script, (vi) neutrality test script, (vii) custom code for data manipulation and visualisation).

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

## Acknowledgements
We thank Peter Morgan and Selvam Thavaraj (King's College London), Iñigo Martincorena and Laura Riva (The Sanger Institute), Mark McGurk (University College London) and the members of the CSCRM for fruitful discussions and comments on the manuscript. We are very grateful to the members of the animal facility, in particular Dionne Cooper, Zoe Bane, and to Matteo Battilochi for assistance with animal experiments, and to Matthew Caley for the anti-laminin-α3 antibody. IS acknowledges the support of a Barts Charity Lectureship (grant MGU045). TAG is grateful for support from Cancer Research UK (A19771). FMW gratefully acknowledges financial support from Cancer Research UK (C219/A23522), the Medical Research Council (G1100073) and the Wellcome Trust (096540/Z/11/Z). We are also grateful for funding from the Department of Health via the National Institute for Health Research Comprehensive Biomedical Research Centre (BRC) award to Guy's & St Thomas' National Health Service Foundation Trust in partnership with King's College London and King's College Hospital NHS Foundation Trust.

## Author contributions
I.S. and F.M.W. designed the study and oversaw the research. I.S., D.A. and F.M.W. designed the experiments. I.S. performed the animal experiments, sample preparation and microdissection, DNA collection, and coordinated all the analyses. M.R. and D.A. coordinated the sequencing pipeline. I.S. and I.T. performed immunohistochemistry on tissue samples and quantification. A.V., M.R. and M.W. wrote analysis codes and performed mathematical and bioinformatic analysis. I.S, A.V., M.R., I.T. and T.A.G. analysed the data. I.S. and F.M.W. wrote the manuscript with input from all the authors.

## Competing interests
F.M.W. is currently on secondment as Executive Chair of the UK Medical Research Council. The remaining authors declare no competing interests.
