## [Peer Review File · Nature Communications]

Reviewers' Comments:

Reviewer #1:

Remarks to the Author:

While this study has innovative aspects, the elements presented in this work have been previously reported. For example, the similarity between 4NQO induced tumors and human disease was assessed (albeit in smaller number of samples) on the mutational, gene expression and epigenetic levels, as well as the dynamics of these changes (PMID: 27447968, 27027432, 26207766, 26110572, 30135512). The findings presented in this study fall short of the purpose of the work and conclusions stated by the authors, as outlined below:

* It is not clear how many animals were used in each arm. The text only mentions the overall number of collected samples (n=69). The number of animals used for 4NQO treatment should be indicated in the first paragraph of the manuscript. Otherwise it is impossible to properly evaluate the significance of the findings.

* It is not clear why authors decided to use FFPE specimens for WES when fresh or fresh frozen samples were available? Dysplastic human samples may not be readily available as FF material, but in 4NQO models plenty of fresh samples are available for the analysis. The neoplastic cells collected by LCM could have substantially improve the data quality and results reported in this manuscript. I would highly encourage the authors to sequence a few FF samples and compare the mutational landscape with that obtained from the FFPE material.

* Authors used fresh ear sample as a control for FFPE derived tongue specimens. Using such analysis the authors discovered over 23K somatic mutations in the "normal" tongue treated with 4NQO. Without sequencing of a fresh tongue, the authors can't rule out the possibility that a substantial number of these mutations are FFPE-induced. Removing mutations with AF <0.10 eliminates all genetic aberrations below 10% frequency, and makes the comparison to human data from TCGA very challenging.

* Mice treated with 4NQO frequently develop synchronous dysplasia/OSCC neoplasms. Why these lesions are not even discussed? Analyzing normal tongue and progressive lesions from the same animal would be crucial for clonal expansion evaluation.

* Several previous studies have reported a low level of Trp53 mutations in 4NQO treated tumors. Also, several studies have used animals with inducible TP53 flox/flox in the oral mucosa to better mimic the human oral cancer, where TP53 was found to be mutated in ~70% of the cases. In this paper, Trp53 was mutated in 30% of the samples (again, knowing the number of animals used is critical for proper evaluation of this study), but the authors do not discuss the differences between their work and previously reported studies.

* All IF and IHC experiments should be quantified based on multiple fields and images should be supplemented with charts.

* The p values for all gene-phenotype association should be clearly labeled in the text. For example, on Figure 4C, it seems that the association between Notch and immune infiltrate has a p value below 0.01 (blue color). However, in the text, it is discussed as significant. In the Supp. Figure 4 however, most Notch associations are indicated as not significant (p=0.019 for immune infiltrate and 0.012 for tumor grade). If the data is not significant, no need to discuss it in the text and definitely it is wrong to call it significant in the text if the numbers don't reach the significance.

* Figure 4e maybe significant, but it is clear that the significance was reached due to a single sample. Due to the low number of Notch mutated specimens the claim that Notch1 mut tumors have higher CD45 infiltration is highly biased.

* To assess the intratumor heterogeneity authors should sequence specimens collected from different areas within the same OSCC. Computational quantification of subclonal selection from single area bulk sequencing data is predictive at the best and was reported multiple times in human OSCC tumors.

Minor comments: Please explain abbreviations the first time they appear in the text

Reviewer #2:

Remarks to the Author:

The manuscript by Sequeria and co-workers addresses the important issue of the effects of smoke on oral SCC genetic landscape. As a model, mice are fed with mutagen Nitroquinoline-oxide and tongues harvested over the time after the end of the treatment for histological and genetic analysis. They found an overlapping between the identified signatures and known human oral and other human SCC types related with tobacco smoking. Genetically, a larger mutational burden was found in more aggressive SCCs as compared to mild dysplasias and hyperplasias. Earlier lesions carry a larger burden of p53/Notch/Fat1 mutations and have larger "immune infiltrates".

The information in the present manuscript are largely descriptive and confirmatory of previous findings. For the identified genes/pools with increased mutation burden, and for the "driver" genes the mechanism is not addressed and the multiple evidence are not definitively investigated at this stage.

Specific points

For WES tongue of Quinoline-treated animals are compared to the ear skin of the same animals. Beside the intrinsic differences present when comparing H&N to skin, it has to be considered that ear skin might also be affected by the treatment as well, and tongue of untreated animals might be used for comparison.

#As they are considered together grades 4 and 5 should be combined also in the graphs of Figure 1

It is not surprising that single nucleotide variation increases with tumor grade and quinoline treatment duration (in hyperplastic lesions) however the authors should address whether these mutations affects function/expression in oral system.

The driver genes represent only a list and no verification for the newly identified (Chuk etc) is made.

Quantifications for the evidence presented in Figure 3 a and b should be provided.

The Authors should provide an explanation why as tumors become more aggressive a lower proliferation rate is observed in Fig4b. The CD45 recognizes all resident and inflammatory-recruited cells; it would be more informative to separate macrophages from leukocytes.

Reviewer #3:

Remarks to the Author:

This is an interesting foundation paper that shows that a mutagen with effects similar to those in cigarette smoke can elicit similar mutations and lesions in the mouse tongue to those recently reported in the human esophagus relative to pack-years and aging. If these events are indeed similar to those underlying more general features of the evolution of oral-pharyngeal squamous cell carcinomas, a murine model will have significant advantages in working out the immunology associated with these cancers in a way that would be more difficult or impossible to assess in patients. The work is well done and presented.

The Notch 1 mutations are interesting as they seem to arise in many of the recent human studies of esophageal epithelia in aging and smoking, and yet seem more common in less advanced lesions than those that progress to invasive cancer. This has raised the question of non-cell autonomous actions of Notch1 mutations, perhaps in altering some form of microenvironment that could ultimately influence progression of cells in a field (authors should also incorporate the paper by Yokoyama and colleagues, *Nature*, 2019). The link the authors make between Notch1 mutations and immune cell infiltrates may relate to this, and the authors should comment on the relative association between Notch1, p53, and stage of lesions to see how they match up with the earlier studies, even if the numbers are low. Further comments on Notch1 are listed at the bottom.

It might also draw attention to the system if the authors could provide some hints as to what can come from the model. Given that the present work was done with FFPE samples with generally low VAF and yet still shows similarities to the human condition is highly encouraging. These findings beg a demonstration of how the epithelial culture models the lab has pioneered might eliminate stromal contaminants and perhaps enable true clonal analyses. Perhaps the other suggestion to improve the visibility of the present approach would be an assessment of matched dysplastic and invasive lesions. The non-invasive dysplastic lesions are presented as very distinct from the invasive lesions with submucosal growth under normal squamous epithelia, but serial sectioning would reveal a dysplastic lesion subtended by invasive disease which could be separately captured to assess their mutational profiles. Again a single example of such a relationship would go far to promote the system.

Other comments:

Fig. 1. b. The text should describe the significance of the regions of the tongue as many will not be familiar with the histology differences if they exist.

Fig. 1. e. Violin plot for grade 4-5 and 6 will not be obvious.

Fig. 4. d. Because the Notch1 mutations were so few, and other Notch1 wt showed significant CD45 infiltration, there will be questions about separating CD45 values independent of Notch1 and examining for other correlations.

Fig. 4. e. The CD45 signal will have to be improved to get the message across. Also efforts to further break down CD45 into hematopoietic lineages would be very helpful not only in the Notch1 mutants but also in the high wt ones.

Response to reviewers of NCOMMS-17-04845

We would like to thank the reviewers for their insightful comments on our manuscript. Here we provide a detailed point-by-point response to each reviewer.

Reviewer #1 (Remarks to the Author):

While this study has innovative aspects, the elements presented in this work have been previously reported. For example, the similarity between 4NQO induced tumors and human disease was assessed (albeit in smaller number of samples) on the mutational, gene expression and epigenetic levels, as well as the dynamics of these changes (PMID: 27447968, 27027432, 26207766, 26110572, 30135512).

We have now cited all of these references. We believe that our study goes well beyond the earlier reports because it provides a comprehensive genomic and clonal analysis of tumours at different stages of development and maps the data onto changes in the tumour microenvironment.

The finding findings presented in this study fall short of the purpose of the work and conclusions stated by the authors, as outlined below:

1* It is not clear how many animals were used in each arm. The text only mentions the overall number of collected samples (n=69). The number of animals used for 4NQO treatment should be indicated in the first paragraph of the manuscript. Otherwise it is impossible to properly evaluate the significance of the findings.

We have included the number of animals in the first paragraph of the Results, as suggested. The number of tumours of each grade is indicated in Figure 1d and we have made more explicit reference to this in the text.

2* It is not clear why authors decided to use FFPE specimens for WES when fresh or fresh frozen samples were available? Dysplastic human samples may not be readily available as FF material, but in 4NQO models plenty of fresh samples are available for the analysis. The neoplastic cells collected by LCM could have substantially improve the data quality and results reported in this manuscript. I would highly encourage the authors to sequence a few FF samples and compare the mutational landscape with that obtained from the FFPE material.

This is an important point, which we now address more fully in the manuscript. We opted for FFPE samples because it is easier to micro-dissect areas of individual tumours, reducing the contamination of the epithelial compartment with adjacent stroma. This is particularly important when analysing small hyperplasias and assessing tumour subclonality.

Several papers (cited) have previously compared FFPE and frozen samples, observing that FFPE variants are of low allelic frequency (<5%), and collectively share a "C>T|G>A" mutational signature known to be an FFPE artefact resulting from cytosine deamination.

The increase in C>T transitions in FFPE samples is most pronounced at CpG dinucleotides. Nevertheless, the error rate, library complexity, enrichment performance, and coverage statistics are not significantly different in fresh and FFPE samples

To take into account the potential artefacts associated with FFPE samples, we used an effective filtering strategy with associated empirical false-discovery estimates in our analysis pipeline. Furthermore, we used FFPE blocks that were <6 months old because artefacts are known to increase with prolonged storage. We performed

rigorous quality control on extracted DNA and assessed sequence data quality, read alignments, library complexity, raw error rate, and consensus base calls. Finally, we confirmed that our samples did not have the typical increase in C>T transitions observed in archival FFPE (see Figure below, which is now incorporated in the manuscript in Supplementary Figure 1c).

3* Authors used fresh ear sample as a control for FFPE derived tongue specimens. Using such analysis the authors discovered over 23K somatic mutations in the “normal” tongue treated with 4NQO. Without sequencing of a fresh tongue, the authors can’t rule out the possibility that a substantial number of these mutations are FFPE-induced. Removing mutations with AF <0.10 eliminates all genetic aberrations below 10% frequency, and makes the comparison to human data from TCGA very challenging.

Formalin-fixed samples can display disproportionate levels of C>T/G>A changes in the 1-10% allele frequency range, as discussed above. We therefore used a filter combining the mutant allele fraction (allele fraction < 0.10) and sequencing coverage (coverage >20). Any mutations that did not pass these criteria were excluded, ruling out the possibility that FFPE-induced mutations were analysed. In addition, we sequenced FFPE derived tongue samples that were histologically normal (see Figure 1d) and had not been treated with 4NQO (MD5547b, MD5548b) and these formed the baseline for analysing 4NQO-induced mutations during tumour progression.

4* Mice treated with 4NQO frequently develop synchronous dysplasia/OSCC neoplasms. Why these lesions are not even discussed? Analyzing normal tongue and progressive lesions from the same animal would be crucial for clonal expansion evaluation.

This is an excellent suggestion. We have now conducted comparative analysis of synchronous lesions in eight animals (new Supplementary Figure 5). In 4 out of 8 animals different lesions had no common mutations, in 2 out of 8 there were ≤3 mutations in common, and in only 1 out of 8 there was a significant number of mutations (n=1189, Supplementary Figure 5a). In this latter case both lesions were located in the dorsal region of the tongue (Supplementary Figure 5b, 2_D_MD5534c and 4_D_MD5534a). We conclude that lesions in different regions of the tongue are not clonally related, but if two lesions are close together they could be clonally related. We have now included this analysis in the manuscript.

5* Several previous studies have reported a low level of Trp53 mutations in 4NQO treated tumors. Also, several studies have used animals with inducible TP53 flox/flox in the oral mucosa to better mimic the human oral cancer, where TP53 was found to be mutated in ~70% of the cases. In this paper, Trp53 was mutated in 30% of the samples (again, knowing the number of animals used is critical for proper evaluation of this study), but the authors do not discuss the differences between their work and previously reported studies.

Vettore et al, 2015 compared the prevalence of mutations in tongue SCC (TSCC) with mutations in total oral cavity (OCSCC). They found Trp53 mutations in >70% of OCSCC, but only in 38.3% of TSCC. Thus our findings are consistent with the earlier study. As we discuss in the text, it is also likely that the lower frequency of Trp53 mutations in our study is due to the fact that we analysed early to late stage tongue lesions, while studies in human mainly involve advanced tumours. We have now revised the text to include the Vettore et al reference.

6* All IF and IHC experiments should be quantified based on multiple fields and images should be supplemented with charts.

This has now been performed. IHC quantification was performed on at least 2 sections/tumour and 3-6 fields/section). We have now added images to show how the quantification was performed (Supplementary Figure 4b). We have also included all the quantification as well as the representative images in Figure 3a-d, Figure 5a-d, Figure 5g-h.

7* The p values for all gene-phenotype association should be clearly labeled in the text. For example, on Figure 4C, it seems that the association between Notch and immune infiltrate has a p value below 0.01 (blue color). However, in the text, it is discussed as significant. In the Supp. Figure 4 however, most Notch associations are indicated as not significant (p=0.019 for immune infiltrate and 0.012 for tumor grade). If the data is not significant, no need to discuss it in the text and definitely it is wrong to call it significant in the text if the numbers don't reach the significance.

We are grateful to the reviewer for pointing out this error, which we have now corrected. In the revised manuscript we have increased the number of sections stained and regions quantified for the CD45⁺ immune infiltrate. Linear regression analysis confirms an association between Notch1 mutations and immune infiltrate (Figure 5e, Supplementary Table 7, $Pr(>|t|) = 0.000696$). We have also verified this correlation by comparing the immune infiltrate in Notch1 wt vs Notch1 mutated samples using an unpaired t-test (Figure 5f, p=0.0083). We have verified all the colours in the heatmap (Figure 5e) and included Supplementary Table 7 to provide all the significant p-values in the linear regression analysis.

8* Figure 4e maybe significant, but it is clear that the significance was reached due to a single sample. Due to the low number of Notch mutated specimens the claim that Notch1 mut tumors have higher CD45 infiltration is highly biased.

In response to this comment we have re-stained the wt and Notch1 mutated tumours for CD45 and enhanced the quantification (see point 7). We still observe a significant increase of CD45⁺ cells in Notch1 mutated tumours (Figure 5e-f, Supplementary Table 7). In addition, we have now stained for other immune cells (T-cells and macrophages). According to the linear regression analysis, Notch1 mutations also correlate with T-cell infiltrate (p=0.02019352, Supplementary Table 7). Thus in the revised manuscript the increased immune infiltrate in Notch1 mutated tumours is confirmed.

9* To assess the intratumor heterogeneity authors should sequence specimens collected from different areas within the same OSCC. Computational quantification of subclonal selection from single area bulk sequencing data is predictive at the best

and was reported multiple times in human OSCC tumors.

While we agree that this would be valuable, the size of the mouse tongue lesions is too small to allow separate analysis of different regions of the same tumour. We have now cited reports of subclonal selection in advanced human OSCC (eg. <https://www.ncbi.nlm.nih.gov/pmc/articles/PMC5369985/>). Notwithstanding any technical limitations of our study, the novelty lies in combining the timeline of tumour evolution with the subclonality analysis. We have been able to compare tumour grade with clonal selection, from early hyperplastic lesions to frank tumours/invasive SCC. While most mild dysplasias present with neutral growth and higher-grade tumours present with patterns that are consistent with clonal mutations, as reported in humans, we have shown that subclones arise as early as grade 3, mild dysplasia (Figure 6c). As we discuss, this could be of prognostic significance in human OSCC.

Minor comments: Please explain abbreviations the first time they appear in the text
This has now been done.

Reviewer #2 (Remarks to the Author):

The manuscript by Sequeria and co-workers addresses the important issue of the effects of smoke on oral SCC genetic landscape. As a model, mice are fed with mutagen Nitroquinoline-oxide and tongues harvested over the time after the end of the treatment for histological and genetic analysis. They found an overlapping between the identified signatures and known human oral and other human SCC types related with tobacco smoking. Genetically, a larger mutational burden was found in more aggressive SCCs as compared to mild dysplasias and hyperplasias. Earlier lesions carry a larger burden of p53/Notch/Fat1 mutations and have larger "immune infiltrates".

The information in the present manuscript are largely descriptive and confirmatory of previous findings. For the identified genes/pools with increased mutation burden, and for the "driver" genes the mechanism is not addressed and the multiple evidence are not definitively investigated at this stage.

We feel that this assessment misses the point of our study. While multiple investigators have looked at the effect of individual genes on cancer progression, our initial goal was to establish whether the mutational signatures of mouse and human OSCC are similar. Having established that they are, we went on to demonstrate how the mutational landscape in the mouse model changes with tumour progression, how particular combinations of mutations have a distinct biological 'fingerprint', and how high genetic heterogeneity may be predictive of early lesions that are likely to progress to more aggressive tumours. We believe that the revised manuscript sets out more clearly the importance of our findings.

Specific points

1# For WES tongue of Quinoline-treated animals are compared to the ear skin of the same animals. Beside the intrinsic differences present when comparing H&N to skin, it has to be considered that ear skin might also be affected by the treatment as well, and tongue of untreated animals might be used for comparison.

Ear biopsies were collected before the start of treatment, ruling out the possibility that ear skin was affected by 4NQO treatment – this point is now clarified in the text. In addition, WES was performed on untreated tongues from two C57Bl/6N control mice age-matched from the same litter (samples MD5547b and MD5548b) following FFPE, demonstrating that FFPE did not introduce artefactual germline mutations

2# As they are considered together grades 4 and 5 should be combined also in the graphs of Figure 1

We have now combined grade 4 and 5 samples in the graphs.

3# It is not surprising that single nucleotide variation increases with tumor grade and quinoline treatment duration (in hyperplastic lesions) however the authors should address whether these mutations affects function/expression in oral system.

To address this, we have now compared the mutations in hyperplastic lesions of mice that had been treated with 4NQO for different lengths of time. See new Figure 4 and accompanying text.

4# The driver genes represent only a list and no verification for the newly identified (Chuk etc) is made.

As explained above, the objective of the study was not to perform mechanistic studies of individual genes. However, previous studies (now cited) have shown that genes involved in the NF- κ B pathway, including Chuk, are upregulated in skin SCCs (<https://mct.aacrjournals.org/content/17/9/2034.long>). We have also cited a very interesting recent report in Science (Loganathan et al, 2020) on how mutations in multiple genes converge on the Notch pathway in HNSCC.

5# Quantifications for the evidence presented in Figure 3 a and b should be provided.

We have now quantified the Lama3a and p53 staining (new Figure 3b-d).

6# The Authors should provide an explanation why as tumors become more aggressive a lower proliferation rate is observed in Fig4b.

We quantified the number of dividing cells (EdU⁺) in the epithelial basal layer of the epithelium. As shown in Supplementary Figure 4b, there is an expansion of undifferentiated cells (Krt14⁺) into the suprabasal layers. Therefore, while there is a decrease in the number of basal EdU⁺ cells through tumour expansion, the total number of dividing cells increases. This is now explained in the text. In addition, although there is no significant difference in proliferation when we compare all groups (one-way ANOVA, non-significant $p=0.1619$), t-tests comparing pairs of tumour grades show a significant increase in proliferation in invasive SCC compared to hyperplasia ($p=0.0262$). The t-tests are now included in Figure 5d.

One-way ANOVA to compare all groups
not significant

Table Analyzed	Proliferation_all_mm2
Data sets analyzed	A-D
ANOVA summary	
F	1.789
P value	0.1619
P value summary	ns
Significant diff. among means (P < 0.05)?	No
R squared	0.1006
Brown-Forsythe test	
F (DFn, DFd)	0.3618 (3, 48)
P value	0.7809
P value summary	ns
Are SDs significantly different (P < 0.05)?	No
Bartlett's test	
Bartlett's statistic (corrected)	1.909
P value	0.5915
P value summary	ns
Are SDs significantly different (P < 0.05)?	No
ANOVA table	
Treatment (between columns)	887132
Residual (within columns)	7934264
Total	8821395
Data summary	
Number of treatments (columns)	4
Number of values (total)	52

t-test analysis between each grade
significant difference only between Invasive SCC and Hyperplasia

Table Analyzed	Proliferation_all_mm2
Column D	Invasive SCC
vs.	vs.
Column A	Hyperplasia
Unpaired t test	
P value	0.0262
P value summary	*
Significantly different (P < 0.05)?	Yes
One- or two-tailed P value?	Two-tailed
t, df	t=2.369, df=24
How big is the difference?	
Mean of column A	1074
Mean of column D	707.4
Difference between means (D - A) ± SEM	-366.4 ± 154.6
95% confidence interval	-685.5 to -47.23
R squared (eta squared)	0.1896

The CD45 recognizes all resident and inflammatory-recruited cells; it would be more informative to separate macrophages from leukocytes.

In light of the reviewer's comments, we have now performed immunostaining and quantification of CD68⁺ macrophages and CD3⁺ T-cells (Figure 5 and Supplementary Figure 5a). We show that while several gene mutations correlate with an enhanced T-cell infiltrate, there is no correlation with macrophage infiltrate (Figure 5e).

Reviewer #3 (Remarks to the Author):

Sequeira et al., Genomic Landscape and Clonal Architecture...

This is an interesting foundation paper that shows that a mutagen with effects similar to those in cigarette smoke can elicit similar mutations and lesions in the mouse tongue to those recently reported in the human esophagus relative to pack-years and aging. If these events are indeed similar to those underlying more general features of the evolution of oral-pharyngeal squamous cell carcinomas, a murine model will have significant advantages in working out the immunology associated with these cancers in a way that would be more difficult or impossible to assess in patients. The work is well done and presented.

The Notch 1 mutations are interesting as they seem to arise in many of the recent human studies of esophageal epithelia in aging and smoking, and yet seem more common in less advanced lesions than those that progress to invasive cancer. This has raised the question of non-cell autonomous actions of Notch1 mutations, perhaps in altering some form of microenvironment that could ultimately influence progression of cells in a field (authors should also incorporate the paper by Yokoyama and colleagues, Nature, 2019).

We now cite this paper, together with the recent paper by Loganathan et al. (2020).

The link the authors make between Notch1 mutations and immune cell infiltrates may

relate to this, and the authors should comment on the relative association between Notch1, p53, and stage of lesions to see how they match up with the earlier studies, even if the numbers are low. Further comments on Notch1 are listed at the bottom.
We have now expanded our analysis and discussion of the link between Notch1 mutations and immune infiltrates.

It might also draw attention to the system if the authors could provide some hints as to what can come from the model. Given that the present work was done with FFPE samples with generally low VAF and yet still shows similarities to the human condition is highly encouraging. These findings beg a demonstration of how the epithelial culture models the lab has pioneered might eliminate stromal contaminants and perhaps enable true clonal analyses.

We have indeed used culture models extensively in our work on normal and malignant epithelial cells. However, the goal of the present study was to integrate the genetic and environmental changes in vivo and for that reason we have not carried out any cell culture experiments.

Perhaps the other suggestion to improve the visibility of the present approach would be an assessment of matched dysplastic and invasive lesions. The non-invasive dysplastic lesions are presented as very distinct from the invasive lesions with submucosal growth under normal squamous epithelia, but serial sectioning would reveal a dysplastic lesion subtended by invasive disease which could be separately captured to assess their mutational profiles. Again a single example of such a relationship would go far to promote the system.

We have acted on this excellent suggestion – see response to point 4 of Reviewer 1, above.

Other comments:

Fig. 1. b. The text should described the significance of the regions of the tongue as many will not be familiar with the histology differences if they exist.

We have included a description of the different tongue regions in Figure 1b, as suggested.

Fig. 1. e. Violin plot for grade 4-5 and 6 will not be obvious.

We have now changed the violin plot into a bar plot, presenting the means, SD and individual data points.

Fig. 4. d. Because the Notch1 mutations were so few, and other Notch1 wt showed significant CD45 infiltration, there will be questions about separating CD45 values independent of Notch1 and examining for other correlations.

As suggested, we have compared the CD45⁺ cells in Notch1 wt samples and have not found another correlation independent of Notch1 mutations (p=0.496).

Predictors	Total number of mutations		
	Estimates	CI	p
	All samples		
	0.13	-0.20 – 0.46	0.444
	Excluding Notch1 mutated samples		
Immune infiltrate CD45 ⁺ cells/mm ²	0.13	-0.23 – 0.49	0.496

Fig. 4. e. The CD45 signal will have to be improved to get the message across. Also efforts to further break down CD45 into hematopoietic lineages would be very helpful not only in the Notch1 mutants but also in the high wt ones.

We have now improved the analysis of immune infiltrates, as discussed in response to Reviewer 2, point 6.

Reviewers' Comments:

Reviewer #1:

None

Reviewer #2:

Remarks to the Author:

The Author have answered skillfully, but fairly, to all the concerns raised in the first reviewing process.
The resubmitted manuscript is therefore significantly improved.

Reviewer #3:

Remarks to the Author:

The authors have done an excellent job at addressing all of my questions.